# The Inhibition of Complement System in Formal and Emerging Indications: Results from Parallel One-Stage Pairwise and Network Meta-Analyses of Clinical Trials and Real-Life Data Studies

**DOI:** 10.3390/biomedicines8090355

**Published:** 2020-09-16

**Authors:** Coralina Bernuy-Guevara, Hassib Chehade, Yannick D. Muller, Julien Vionnet, François Cachat, Gabriella Guzzo, Carlos Ochoa-Sangrador, F. Javier Álvarez, Daniel Teta, Débora Martín-García, Marcel Adler, Félix J. de Paz, Frank Lizaraso-Soto, Manuel Pascual, Francisco Herrera-Gómez

**Affiliations:** 1Pharmacological Big Data Laboratory, University of Valladolid, 47005 Valladolid, Spain; coralber@gmail.com (C.B.-G.); alvarez@med.uva.es (F.J.Á.); de_paz@med.uva.es (F.J.d.P.); flizarasos@usmp.pe (F.L.-S.); 2Pediatric Nephrology Unit, Lausanne University Hospital and University of Lausanne, 1100 Lausanne, Switzerland; hassib.chehade@chuv.ch (H.C.); Francois.Cachat@chuv.ch (F.C.); 3Transplantation Center, Lausanne University Hospital and University of Lausanne, 1100 Lausanne, Switzerland; yannick.muller@unige.ch (Y.D.M.); julien.vionnet@kcl.ac.uk (J.V.); Gabriella.Guzzo@chuv.ch (G.G.); Manuel.Pascual@chuv.ch (M.P.); 4King’s College London, London WC2R 2LS, UK; 5Clinical Epidemiology Support Office, Sanidad de Castilla y León, 49022 Zamora, Spain; cochoas2@gmail.com; 6Ethics Committee of Drug Research–east Valladolid area, University Clinical Hospital of Valladolid, 47005 Valladolid, Spain; 7Department of Nephrology, Hôpital du Valais, 1950 Sion, Switzerland; daniel.teta@hopitalvs.ch; 8Clinical Nephrology Unit, University Clinical Hospital of Valladolid, 47003 Valladolid, Spain; deboramarg@yahoo.es; 9Center for Medical Oncology & Hematology, Hospital Thun, 3600 Thun, Switzerland; marcel.adler@spitalstsag.ch; 10Centro de Investigación en Salud Pública, Instituto de Investigación de la Facultad de Medicina Humana, Universidad de San Martín de Porres, Lima 15024, Peru; 11Department of Nephrology, Hospital Virgen de la Concha, 49022 Zamora, Spain; 12Castile and León’s Research Consolidated Unit n° 299, 47011 Valladolid, Spain

**Keywords:** complement inactivating agents, meta-analysis as topic, biological products

## Abstract

This manuscript presents quantitative findings on the actual effectiveness of terminal complement component 5 (C5) inhibitors and complement component 1 (C1) esterase inhibitors through their formal and common “off-label” (compassionate) indications. The results emanated from pairwise and network meta-analyses to present evidence until September 2019. Clinical trials (CT) and real-life non-randomized studies of the effects of interventions (NRSI) are consistent on the benefits of C5 inhibitors and of the absence of effects of C1 esterase inhibitors (*n* = 7484): Mathematically, eculizumab (surface under the cumulative ranking area (SUCRA) >0.6) and ravulizumab (SUCRA ≥ 0.7) were similar in terms of their protective effect on hemolysis in paroxysmal nocturnal hemoglobinuria (PNH), thrombotic microangiopathy (TMA) in atypical hemolytic uremic syndrome (aHUS), and acute kidney injury (AKI) in aHUS, in comparison to pre-/off-treatment state and/or placebo (SUCRA < 0.01), and eculizumab was efficacious on thrombotic events in PNH (odds ratio (OR)/95% confidence interval (95% CI) in CT and real-life NRSI, 0.07/0.03 to 0.19, 0.24/0.17 to 0.33) and chronic kidney disease (CKD) occurrence/progression in PNH (0.31/0.10 to 0.97, 0.66/0.44 to 0.98). In addition, meta-analysis on clinical trials shows that eculizumab mitigates a refractory generalized myasthenia gravis (rgMG) crisis (0.29/0.13 to 0.61) and prevents new acute antibody-mediated rejection (AMR) episodes in kidney transplant recipients (0.25/0.13 to 0.49). The update of findings from this meta-analysis will be useful to promote a better use of complement inhibitors, and to achieve personalization of treatments with this class of drugs.

## 1. Introduction

The complement system that functions to protect the host against infection, principally by opsonizing and lysing pathogens, can cause tissue damage and inflammation in the cases of dysregulation that occurs commonly due to rare defects genetically originated [1]. In this sense, a spectrum of diseases and patients benefit from pharmacological inhibition of various complement components. The inhibition of terminal complement component 5 (C5) is approved by the Food and Drug Administration (FDA)- and/or the European Medicine Agency (EMA) for the treatment of paroxysmal nocturnal hemoglobinuria (PNH), atypical hemolytic uremic syndrome (aHUS), and more recently to treat refractory generalized myasthenia gravis (rgMG). The two currently available C5 inhibitors, eculizumab and ravulizumab, are also used in the field of solid organ transplantation to treat acute antibody-mediated rejection (aAMR) and delayed graft function (DGF). Eculizumab and ravulizumab are used regardless of their well-known chemical differences, i.e., the four aminoacid substitutions for pH-dependent binding to increase the natural Fc receptor recycling of ravulizumab [2], but without knowledge of more advantages other than a longer half-life of ravulizumab as compared to eculizumab [2].

In addition to C5 inhibitors, various complement component 1 (C1) esterase inhibitors (e.g., Berinert^®^, Cinryze^®^, Haegarda^®^, Ruconest^®^) that are used in the treatment of hereditary angioedema (HAE) have been proposed as a possible alternative to eculizumab in the treatment/prevention of aAMR and DGF [3]. C1 esterase inhibitors act upstream in the complement classical pathway but also in the contact pathway and in decreasing the liberation of bradykinin [4].

### Hypothesis and Study Objective

Targeting the complement, particularly C5, has gained a renewed interest in the last 10 years, often in an off-label manner, for severe conditions lacking any curative treatment [5,6,7]. Indeed, to date, the number of “off-label” or compassionate indications for these drugs can exceed the “officially” approved indications, and this difference might even increase in the future [8].

Importantly, as seen in other areas [9], the actual effectiveness of complement inhibition and differences between each product (or dosing scheme) can only be correctly evaluated if both clinical trials and observational studies assessing ‘real-life’ patients suffering from diseases associated with complement activation are evaluated. Innovative designs should be conceived to carry out systematic reviews to encompass all available evidence both on clinical trials and real-life studies, and network meta-analysis by using standardized tools should be performed to calculate the individual effects of each drug [10].

This report presents separate parallel one-stage pairwise and network meta-analyses on clinical trials and real-life non-randomized studies of the effects of interventions (NRSI) to address the effectiveness of complement inhibition in the treatment of PNH, aHUS, and rgMG (approved indications), and in the treatment/prevention of aAMR and DGF after solid organ transplantation (“off-label” indications).

## 2. Materials and Methods

Mathematical and non-mathematical results presented here respond to the review question on how efficacious are C5 inhibitors and C1 esterase inhibitors in the treatment of PNH, aHUS, and rgMG, and in the treatment/prevention of aAMR and DGF (Table 1). Literature search, study screening and selection, and data extraction are reported in details in our registered protocol in the International prospective register of systematic reviews PROSPERO, created on 21 October 2019, and updated on 3 February 2020 (CRD42019130690, https://www.crd.york.ac.uk/prospero/display_record.php?ID=CRD42019130690). For the interest of readers, search strategy formulation is available online (Panel S1).

Statistical analysis was carried out on aggregate data, after assessing risk of bias in both clinical trials [11] and real-life NRSI [12]. Pooled ORs and 95% CIs for treatment failure outcomes in PNH (hemolysis, thrombosis, CKD apparition/progression), aHUS (TMA, AKI), rgMG, and aAMR, prevention failure in AMR, and DGF were obtained via pairwise meta-analysis (Mantel–Haenszel random-effect method), after verifying heterogeneity (χ^2^, I^2^) and the presence of reporting bias (visual inspection of funnel plots and calculation of Egger’s test, if necessary), using Review Manager software (RevMan) version 5.3 (Cochrane Collaboration) and META-analysis package FOr R (METAFOR) version 2.4 (R project). Pooled ORs and 95% CrIs for the same outcomes evaluated at the pairwise level were calculated via Bayesian network meta-analysis (Markov chain Monte Carlo simulation on vague priors random-effect method for ‘bad’ outcomes and zero values correction), with calculation of the surface under the cumulative ranking area (SUCRA) corresponding to drugs/schemes described in the included studies, after verifying convergence (Brooks–Gelman–Rubin method) and inconsistency, using NetMetaXL software (Canadian Agency for Drugs and Technologies in Health and Cornerstone Research Group) [13].

Skewed and non-quantitative data were presented descriptively following the recommendations of the Centre for Reviews and Dissemination (University of York) [14]. As previously made in other meta-analysis from our team [10], a multidisciplinary supervision mechanism for the contextualization of findings from this summary was planned, with specialists in nephrology (D.M.-G. and D.T.), hematology (M.A.), immunology (J.V.), epidemiology (F.J.d.P. and F.L.-S.), and translational pharmacology (F.J.A.).

## 3. Results

### 3.1. Systematic Narrative Synthesis

The findings presented here are reported in accordance to the Preferred Reporting Items for Systematic reviews and Meta-Analyses (PRISMA) recommendations [15], and strictly adhere to the PRISMA extension statement for reporting of systematic reviews incorporating network meta-analyses of healthcare interventions [16].

Clinical trials and real-life NRSI were searched separately in a parallel one-stage selection procedure, as depicted in Figure 1 (see Materials and Methods for further details). After excluding clearly irrelevant studies (i.e., pre-clinical studies and clinical trials with no evaluation of the eligible outcomes, other observational studies that did not meet the conditions to be considered as real-life NRSI, such as case series or case reports), 28 pharmaceutical industry-sponsored clinical trials corresponding to phases 1 to 3 evaluation of various complement inhibitors, and 15 real-life NRSI reflecting uses of these medicines in real-world settings were found to be eligible: these studies assess outcomes in PNH (No. of clinical trials/real-life NRSI: 7/7), aHUS (7/8), rgMG (3/0), aAMR (6/0), and DGF (5/0), and included a total population of 7484 participants [17,18,19,20,21,22,23,24,25,26,27,28,29,30,31,32,33,34,35,36,37,38,39,40,41,42,43,44,45,46,47,48,49,50,51,52,53,54,55,56,57,58,59,60,61,62,63,64,65,66,67,68,69,70,71,72,73,74,75,76,77,78,79,80,81,82,83,84,85,86,87,88,89,90,91].

Demographics and clinical details of study participants and the characteristics of the included studies are available for readers online (Appendix A). Results from our meta-analytic calculations involved 93.3% of the total number of study participants and cover the inhibition of C5 protein (*n* = 3045), the inhibition of C1 esterase (*n* = 105), and the comparisons for these interventions (*n* = 4334) (Table 2).

All real-life NRSI and all clinical trials but one randomized study [91] were published in peer-reviewed journals. In most cases, data of a single study were displayed in more than one published article. Oral communications and posters presented in meetings of medical societies involved in the treatment of the diseases addressed in this meta-analysis contain also critical information and can be part of the included studies [52,56,64,74,84,85].

In 15 clinical trials (number of trials in PNH/aHUS/rgMG/aAMR: 4/7/1/3, respectively), inclusion of participants into the study was not followed by random allocation of such individuals into intervention and control groups. These single-arm trials used pre- and/or off-treatment state [17,18,20,21,22,23,24,25,45,46,47,48,49,50,51,52,79] or historical cohorts [80,82,87] as comparators for the evaluation of complement inhibition effectiveness.

Randomization was performed in trials in PNH (number of placebo-/standard of care (SOC)-/active-controlled trials: 1/0/2) [19,26,27], in trials assessing eculizumab as treatment of rgMG crises (2/0/0) [77,78], as well as in trials assessing complement inhibitors, respectively, as treatment (1/0/0) or prevention of new aAMR episodes (1/1/0) in kidney recipients [81,83,84,85,86], and as prevention of DGF after kidney transplantation (5/0/0) [88,89,90,91].

Importantly, all participants in non-randomized single-arm trials and all those undergoing placebo or the intervention (complement inhibition) in randomized two-arm trials did not stop rescue treatments (e.g., plasma exchanges and/or intravenous immunoglobulin infusion for treating aHUS, aAMR, and rgMG, immunosuppressive schemes for preventing DGF), nor maintenance treatments (e.g., erythropoietin, corticosteroids and anticoagulants in PNH, maintenance immunosuppression in transplant recipients).

Real-life NRSI comprised most analyzed cases (*n* = 5279), i.e., approximately two-thirds, while the remaining were participants in clinical trials (*n* = 2205). The comparators in real-life NRSI were pre-eculizumab era individuals, i.e., patients who never underwent complement inhibition (56%), and individuals treated with complement inhibitors in their off-treatment state, i.e., patients who discontinued complement inhibitors for various reasons (33%), and patients in their pre-treatment state, i.e., before receiving complement inhibition (11%).

Furthermore, clinical trials and real-life NRSI did not assess adult and pediatric populations separately [17,18,19,20,21,22,23,24,25,26,27,28,29,30,31,32,33,34,35,36,37,38,39,40,41,42,43,44,45,46,47,48,49,50,51,52,53,54,55,56,57,58,59,60,61,62,63,64,65,66,67,68,69,70,71,72,73,74,75,76]. In one trial in aAMR [83] and in two trials in DGF [88,89], DGF prevention and aAMR prevention were not pre-specified as study outcomes, suggesting a potential high risk of alpha inflation (false discovery rate). Results from our evaluation of risk of bias in included studies are available for readers online (Appendix A).

### 3.2. Quantitative Analysis

Thirteen non-randomized single-arm trials and one randomized two-arms trial provided numerical data from individuals with PNH (*n* = 751) and aHUS (*n* = 322), that were entered in multiple-treatments’ meta-analysis calculations. As depicted in Figure 2, the Bayesian network diagrams corresponding to these analyses illustrates the scarcity of available evidence.

Nevertheless, as presented in Figure 3, summary estimates on, respectively, hemolysis in PNH [17,18,19,20,21,22,23,24,25,26,27], thrombotic microangiopathy (TMA) in aHUS [45,46,47,48,49,50,51,52], and acute kidney injury (AKI) in aHUS [45,46,47,48,49,50,51,52], demonstrate a significant protective effect of eculizumab (odds ratio (OR)/95% credible interval (95% CrI): 0.03/0.00 to 0.21, 0.13/0.04 to 0.44, 0.01/0.00 to 0.07) and ravulizumab (0.02/0.00 to 0.29, 0.08/0.01 to 0.61, 0.02/0.00 to 0.34) compared to pre-/off-treatment state and/or placebo (which including rescue/maintenance treatments).

Importantly, with regard to hemolysis in PNH, taking into account only trials using pre-/off-treatment state as comparators (i.e., by excluding the only randomized placebo-controlled trial available), the protective effect of eculizumab (0.01/0.00 to 0.04), and ravulizumab (0.01/0.00 to 0.06) persisted.

On the basis of the surface under the cumulative ranking area (SUCRA), eculizumab (>0.6) and ravulizumab (≥0.7) were similar in terms of their effects on the above mentioned three outcomes, and a markedly difference between treat and not to treat with C5 inhibitors (<0.01) was observed (Table 3). Vague prior random-effects heterogeneity in these calculations was in part counterbalanced by the absence of inconsistency (Appendix A).

Fifteen real-life NRSI including individuals with PNH (*n* = 4189) and aHUS (*n* = 1090) were assessed mathematically into pairwise level. As observed in Figure 4, a protective effect from eculizumab was evident in hemolysis in PNH [28,29,30,31,32,33,34,35,36,37,38,39,40,41,42,43,44], TMA in aHUS [53,54,55,56,57,58,59,60,61,62,63,64,65,66,67,68,69,70,71,72,73,74,75,76], and AKI in aHUS [53,54,55,56,57,58,59,60,61,62,63,64,65,66,67,68,69,70,71,72,73,74,75,76] according to summary estimates obtained (OR/95% confidence interval (95% CI): 0.15/0.08 to 0.28, 0.16/0.06 to 0.46, 0.27/0.18 to 0.42). Considerable heterogeneity (I^2^ >80%) and funnel plot asymmetry affected effect estimates, particularly those corresponding to hemolysis in PNH (Egger’s test (t)/degrees of freedom (df)/p: −2.8186, 17, 0.0372) and TMA in aHUS (−2.3591, 13, 0.0414).

As shown in Figure 5, eculizumab had a positive effect, respectively, in clinical trials and real-life NRSI, on thrombotic events in PNH (0.07/0.03 to 0.19, 0.24/0.17 to 0.33) and chronic kidney disease (CKD) occurrence/progression in PNH (0.31/0.10 to 0.97, 0.66/0.44 to 0.98). No (or non-important) heterogeneity and almost no (or weak) funnel plot asymmetry affected these calculations.

Again at the pairwise level, when combining data from, respectively, the two randomized trials in rgMG and the extension follow-up study of one of them [76,77,78], and the four trials in aAMR prevention [82,86,87,88], eculizumab had a positive effect (0.29/0.13 to 0.61, 0.25/0.13 to 0.49). As shown in Figure 6, although the overall effect of complement inhibition regarding the prevention of new aAMR episodes in kidney recipients was protective, the absence of an effect from the inhibition of C1 esterase allows for seeing an effect only from eculizumab (0.24/0.10 to 0.56) [83,84,85,89], suggesting the efficacy of targeting C5. Contrarily, no effect from complement inhibitors on the prevention of DGF was observed [83,88,89,90,91]. Heterogeneity (43%) affected summary estimates in subgroup analysis of eculizumab to prevent aAMR, and funnel plot asymmetry was particularly noted in the analysis on prevention of DGF. No effect from the inhibition of C1 esterase for treating aAMR [80,81] was found (data not shown).

## 4. Discussion

### 4.1. Important Messages

This manuscript presented gathers and evidence on the effectiveness of complement inhibition. Formal and common “off-label” (compassionate) indications of these medicines are covered. Our results include a total of 7484 participants and confirmed that C5 inhibitors are effective (i) to treat PNH, aHUS, and rgMG crises, and (ii) to prevent aAMR episodes. Clinical trials and real-life NRSI are consistent on the beneficial effect of C5 inhibition. The two available C5 inhibitors eculizumab and ravulizumab are similar regarding their effect. The evidence on the inhibition of C1 esterase is still scarce, and data from our analysis showed no effects.

Complement inhibition has become in the last decade a new therapeutic option for a number of rare diseases, most of them leading to death, and that constitute a hard societal burden all over the world [5,6,7]. In some of these indications (such as PNH), it is important to emphasize that C5 inhibition provides only symptomatic relief. Over the past decade, there has been an impressive increase in our comprehension of the role of complement in diverse physiological and pathophysiological states. Together with the discovery of complement inhibitors, assessing both clinical trials and real-life studies, it may allow an optimal use of these drugs. This is the first summary on the benefits of complement inhibitors in various conditions, after verifying that mathematical dichotomous data on various indications for these drugs were available, and with the intention to go beyond clinical trial evidence.

The effect of complement inhibition on several key outcomes is reported. This effect persists in RCT and also in real-life NRSI, despite the fact that the quality of the included individual studies greatly differs, and the existence of statistical heterogeneity and reporting bias in many studies. In particular, with regard to differences between participants in different latitudes leading to heterogeneity, the genetic background should be considered, among other factors.

Moreover, separate summary effect estimates for eculizumab and ravulizumab are presented only for PNH and aHUS. In these two disorders, eculizumab was found to be similar to ravulizumab in terms of their protective effect. Evidence shows thus that eculizumab and ravulizumab play an important role in the management of PNH and aHUS. In our opinion, decisions to use one or the other C5 inhibitor should be based on other factors such as costs, insurance coverage, availability, local expertise, etc. In addition, in these and other indications, long-term use of complement inhibitors remains to be evaluated.

Finally, very few publications investigating the effect of C1 esterase inhibitors were included, and no effects on various pre-defined outcomes were found.

### 4.2. Findings in Context

The most recent summaries on complement inhibition have not addressed the impact of these drugs from a pharmacoepidemiological perspective, that is, for the moment, there are no comprehensive assessments on the benefits of using these medications through their multiple indications [92,93]. However, the evidence body is still small enough to cover mathematically all complement activation diseases.

Our pharmacometrical assessement of complement inhibition clearly showed a significant benefit of treating patients (compared to placebo or historical cohort or standard-of-care), and this is the main finding of our research. Furthermore, we found no difference between eculizumab and ravulizumab.

As demonstrated in a previous work performed by our team, staged systematic review processes and network meta-analysis assessments lead to a more exhaustive evaluation of evidence [10], particularly if the diseases/conditions studied are rare. Updating regularly systematic reviews and meta-analyses is also another very important aspect to take into account [94]. In this sense, the findings presented here should not be interpreted as definitive or categorical: results from a new 26-week, single arm, open-label, phase 3 study of ravulizumab in aHUS, called the 312 study (NCT03131219), will be available soon [95].

### 4.3. Study Limitations

There are some limitations that should be mentioned. Overall, the entire body of evidence is small, even if our summary included real-life studies supporting clinical trials [9]. Analyses performed have attempted to present findings as free as possible from heterogeneity and reporting bias, the common limitations in meta-analysis [96,97]. Such limitations may constitute a discouraging finding, particularly into calculations with real-life NRSI. Nevertheless, although calculations with clinical trials were less affected by heterogeneity, clinical trials were mostly non-randomized.

Furthermore, non-negligible limits by addressing several and different diseases/conditions is to lose nuance in the discussion of actual benefits from complementary inhibition across such affections. For instance, in the setting of transplantation, the various existing types of antibody-mediated rejection should be taken into account: C5 inhibitors do not have a proven effect on chronic AMR [98], which may be interpreted as inconsistent regarding the effect on aAMR supporting our findings. The fact that evidence is available only on kidney transplantation should also be considered, as effectiveness of complement inhibition may vary in recipients of other organs.

### 4.4. Future Research

Head-to-head comparison between eculizumab and ravulizumab only exists for PNH and aHUS. The effect of eculizumab and ravulizumab in other complement-mediated disorders should be investigated in adequately designed RCTs, and their safety profile carefully compared. In addition, future studies should investigate the effect of complement inhibition in other diseases such as neuromyelitis optica or Guillain–Barre disease, for instance. Finally, the effect of C1 esterase inhibitors should be explored further, as the data currently available remain scarce and the potential role of C1 esterase inhibitors cannot be assessed properly at this stage.

### 4.5. Conclusions and Regulatory Considerations

Clinical trials and real-life studies support a beneficial effect of C5 inhibitors in the treatment of PNH, aHUS and rgMG crises, and the prevention of new aAMR episodes in kidney recipients. Available evidence that certainly involve C5 inhibitors made it clear that it is better to treat (SUCRA > 0.6) than not to treat (SUCRA < 0.01), and there is no apparent difference between eculizumab and ravulizumab.

Stronger evidence on beneficial effects in PNH, aHUS and rgMG, as well as in acute antibody-mediated rejection of kidney transplants, is currently available for C5 inhibitors, as compared to C1 esterase inhibitors. Updating this evidence (at regular intervals) is thus important, particularly to promote a better use of complement inhibitors, and to achieve personalization of treatments with this class of drugs.

## Figures and Tables

**Figure 1 biomedicines-08-00355-f001:**
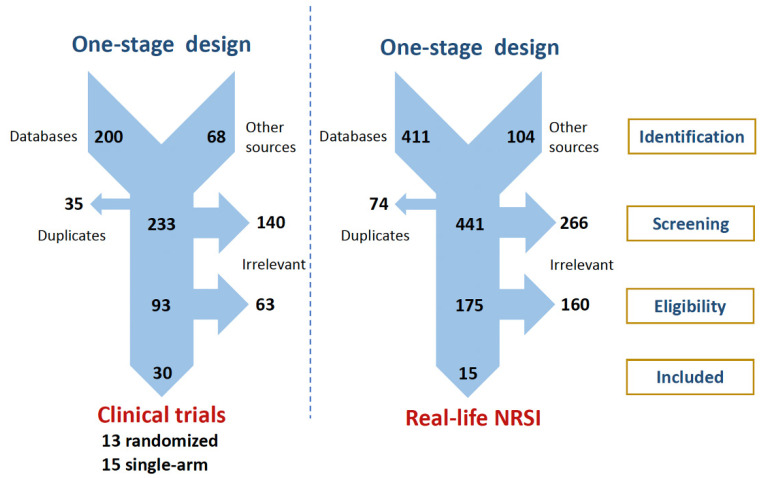
PRISMA flowcharts presenting our parallel one-stage systematic review selection process for retrieving complement inhibition evidence on clinical trials and real-life NRSI. NRSI, non-randomized studies of the effects of interventions; PRISMA, Preferred Reporting Items for Systematic Reviews and Meta-Analyses.

**Figure 2 biomedicines-08-00355-f002:**
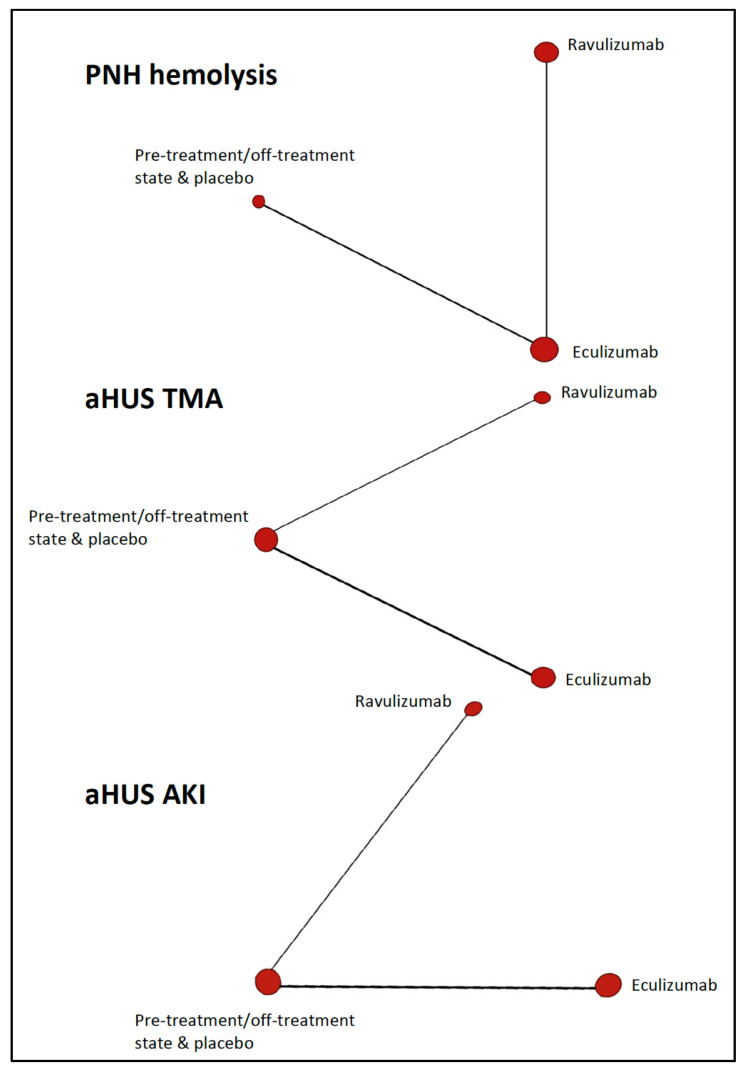
Bayesian network diagrams for the competing complement C5 inhibitors corresponding to three outcomes (clinical trials). aHUS, atypical hemolytic uremic syndrome; AKI, acute kidney injury; PNH, paroxysmal nocturnal hemoglobinuria; TMA, thrombotic microangiopathy.

**Figure 3 biomedicines-08-00355-f003:**
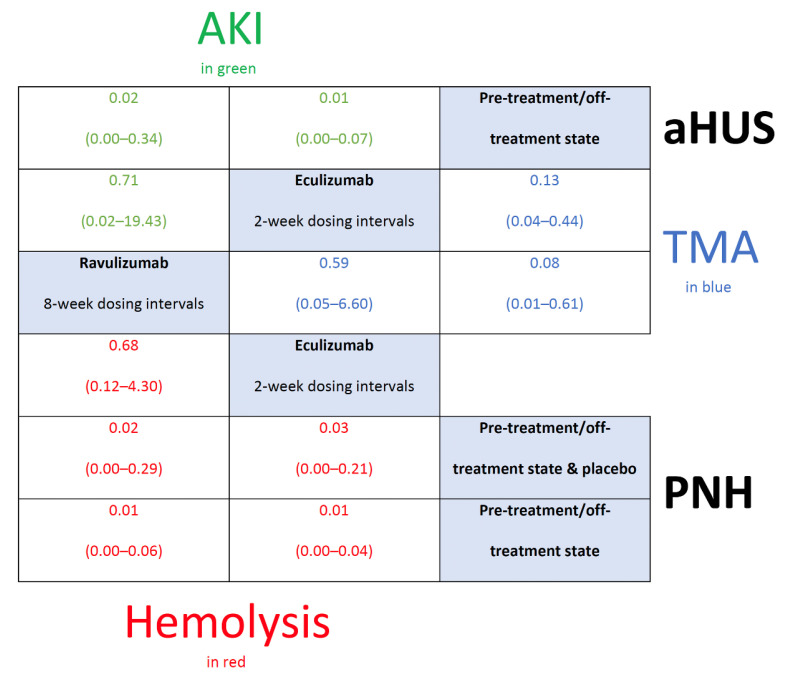
League table showing effect estimates of the assessed complement C5 inhibitors (eculizumab vs. ravulizumab) against the comparators on three outcomes (acute kidney injury (green), thrombotic microangiopathy (blue), and hemolysis (red)) (clinical trials). aHUS, atypical hemolytic uremic syndrome; AKI, acute kidney injury; PNH, paroxysmal nocturnal hemoglobinuria; TMA, thrombotic microangiopathy.

**Figure 4 biomedicines-08-00355-f004:**
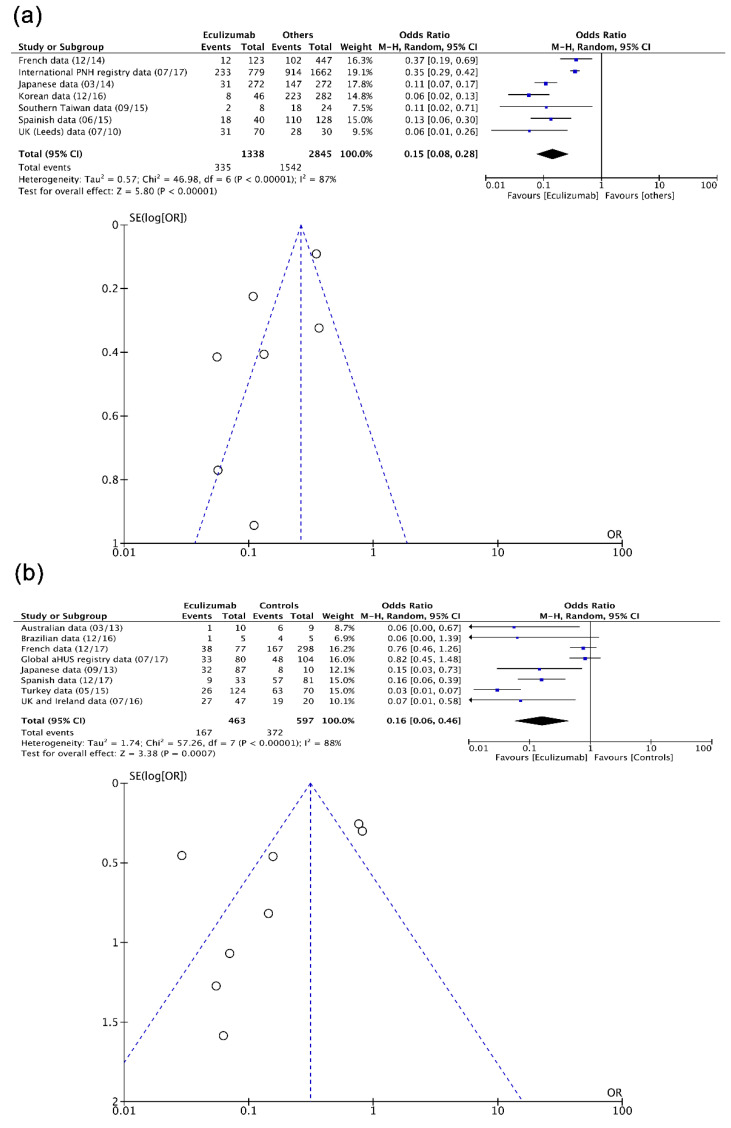
Forest and funnel plots showing effect estimates of eculizumab (real-life NRSI) in (**a**) hemolysis in PNH, (**b**) TMA in aHUS, and (**c**) AKI in aHUS. aHUS, atypical hemolytic uremic syndrome; AKI, acute kidney injury; CI, confidence interval; M–H, Mantel–Haenszel test; NRSI, non-randomized studies of the effects of interventions; PNH, paroxysmal nocturnal hemoglobinuria; SE, standard error; TMA, thrombotic microangiopathy.

**Figure 5 biomedicines-08-00355-f005:**
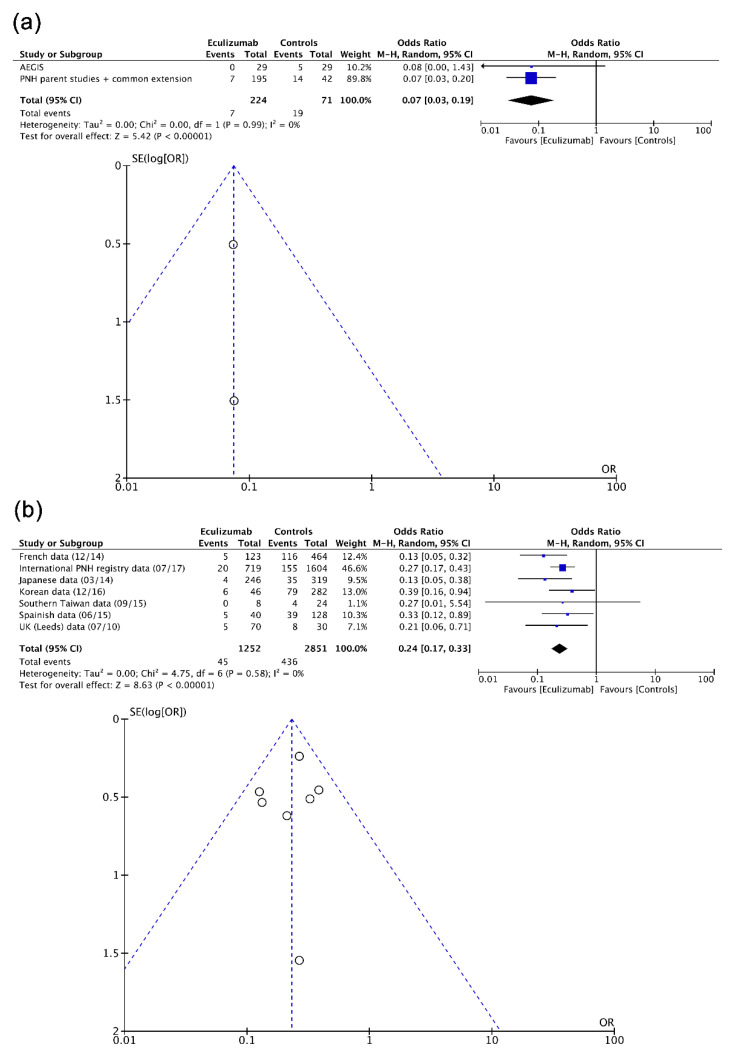
Forest and funnel plots showing effect estimates of eculizumab for (**a**) clinical trials and (**b**) real-life NRSI in thrombotic events in PNH, and for (**c**) clinical trials and (**d**) real-life NRSI in CKD apparition/progression in PNH. CI, confidence interval; CKD, chronic kidney disease; M–H, Mantel–Haenszel test; NRSI, non-randomized studies of the effects of interventions; PNH, paroxysmal nocturnal hemoglobinuria; SE, standard error.

**Figure 6 biomedicines-08-00355-f006:**
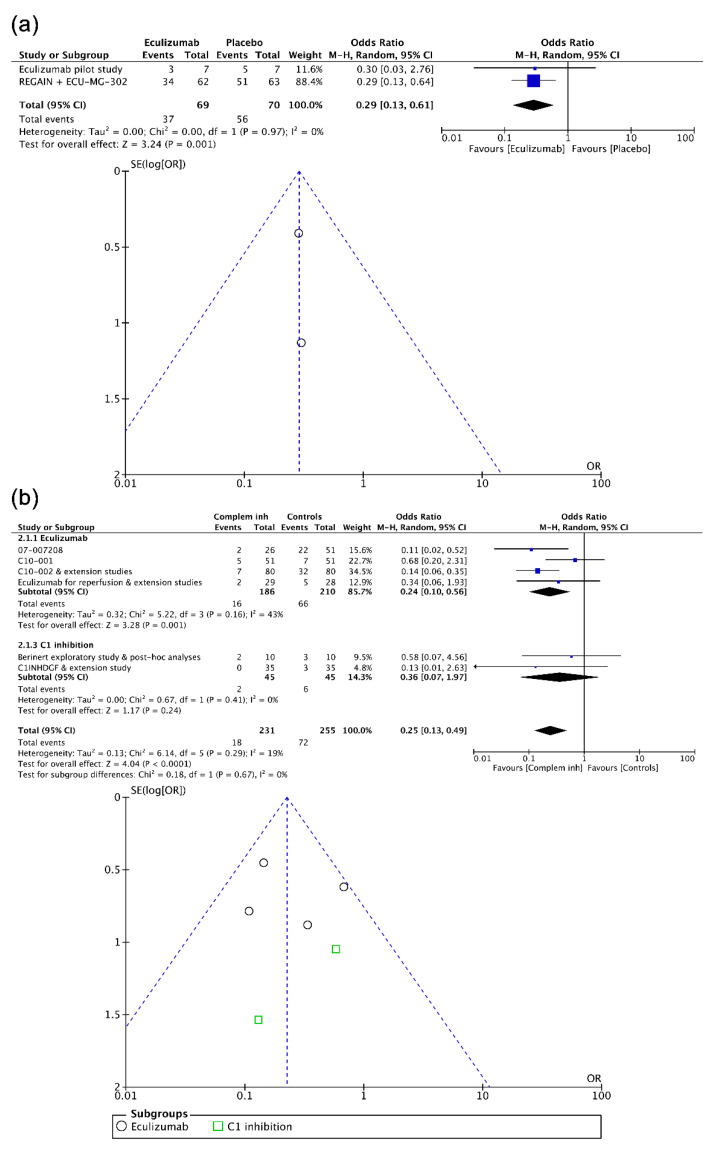
Forest and funnel plots showing effect estimates of complement inhibition (clinical trials) corresponding to (**a**) the treatment of rgMG crises, (**b**) the prevention of new acute AMR episodes, and (**c**) the prevention of DGF after kidney transplantation. AMR, antibody-mediated prevention; CI, confidence interval; M–H, Mantel–Haenszel test; PBO, placebo; rgMG, refractory generalized myasthenia gravis; SE, standard error.

**Table 1 biomedicines-08-00355-t001:** Study eligibility.

	Criteria
Participants	Adult and pediatric individuals affected by or at higher risk of developing PNH attacks, aHUS, rgMG, aAMR episodes, or DGF.
Interventions	Commercial C5 inhibitors (e.g., eculizumab, ravulizumab) and C1-inhibitors (e.g., Berinert^®^, Cinryze^®^, Haegarda^®^, Ruconest^®^).
Comparisons	Placebo, pre-/off-treatment state, historical cohorts that did not receive the interventions, and any other therapeutic strategy (e.g., SOC) including active drugs when it was considered as comparators in the eligible studies.
Type of study	RCT including their extension follow-up studies/post-hoc analyses, in addition to historically controlled interventional studies and other non-randomized (single arm) clinical trials.
	Real-life NRSI (e.g., registry studies and other real-world data studies).

Abbreviations: aHUS, atypical hemolytic uremic syndrome; aAMR, acute antibody-mediated rejection; DGF, delayed graft function; NRSI, non-randomized studies of the effects of interventions; PNH, paroxysmal nocturnal hemoglobinuria; RCT, randomized controlled trial; rgMG, refractory generalized myasthenia gravis; SOC, standard of care.

**Table 2 biomedicines-08-00355-t002:** Number of participants from studies for which numerical data were available for analysis.

	Clinical Trials			Real-Life NRSI		
	C5 Inhibition	C1 Inhibition	Controls	C5 Inhibition	C1 Inhibition	Controls
PNH	665	-	86	1338	-	2851
aHUS	137	-	185	463	-	627
rgMG	69	-	70	-	-	-
aAMR	186	60	285	-	-	-
DGF	187	45	230	-	-	-
Total	1244	105	856	1801	-	3478

Abbreviations: aAMR, acute antibody-mediated rejection; aHUS, atypical hemolytic uremic syndrome; DGF, delayed graft function; NRSI, non-randomized studies of the effects of interventions; PNH, paroxysmal nocturnal hemoglobinuria; rgMG, refractory generalized myasthenia gravis.

**Table 3 biomedicines-08-00355-t003:** SUCRA-based ranking of C5 inhibitors evaluated (clinical trials).

Drug Intervention ^†^ C5 Inhibitors	SUCRA ^‡^ Outcomes: A/B/C ^§^
Eculizumab	0.637/0.642/0.797
Ravulizumab	0.860/0.850/0.700
Pre-treatment/off-treatment states or placebo	0.002/0.007/0.003

^§^ Hemolysis (**A**) in PNH, and TMA (**B**) and AKI (**C**) in aHUS, were the outcomes assessed into network level. ^†^ The two commercial C5 inhibitors analyzed were ranked according to probabilities for being the best, the second best, the third best, and so on P(v=b), b=1,…,a following Markov chain Monte Carlo methods. ^‡^ SUCRA for each C5 inhibitor v out of the a competing C5 inhibitors requires calculation of the a vector of the cumulative probabilities $cum_{v,b}$ to be among the b best drug, b=1,…,a. Abbreviations: aHUS, atypical hemolytic uremic syndrome; AKI, acute kidney injury; PNH, paroxysmal nocturnal hemoglobinuria; SUCRA, surface under the cumulative ranking area; TMA, thrombotic microangiopathy.

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
