# Peer review of "The Inhibition of Complement System in Formal and Emerging Indications: Results from Parallel One-Stage Pairwise and Network Meta-Analyses of Clinical Trials and Real-Life Data Studies"

_biomedicines, 2020, doi:10.3390/biomedicines8090355_

Round 1

Reviewer 1 Report

The manuscript provides a very detailed meta-analysis on the use of a variety of complement inhibitors in the treatment of rare diseases. The effectiveness of C5 inhibition in PNH is very strong, but less so in other disease types and other therapeutic agents.

My only major criticism is that the text in the tabular portions in figures 4, 5 and 6 is very small and difficult to read. This needs to be addressed.

Line 226. It is important to emphasize that C5 inhibition provides symptomatic relief for PNH and can in no way be seen as a curative treatment.

Author Response

The manuscript provides a very detailed meta-analysis on the use of a variety of complement inhibitors in the treatment of rare diseases. The effectiveness of C5 inhibition in PNH is very strong, but less so in other disease types and other therapeutic agents.

My only major criticism is that the text in the tabular portions in figures 4, 5 and 6 is very small and difficult to read. This needs to be addressed.

Thank you very much for your comments. We agree with the fact the tabular portions of Figure 4, 5 and 6 were difficult to read. These portions have now been enlarged in accordance to your suggestion.

Line 226. It is important to emphasize that C5 inhibition provides symptomatic relief for PNH and can in no way be seen as a curative treatment.

We thank you for raising this important point. The text has now been amended (Discussion, subsection “Important message”, second paragraph, lines 235 to 239) and states:

“Complement inhibition has become in the last decade a new therapeutic option for a number of rare diseases, most of them leading to death, and associated with a heavy societal burden [5–7]. In some of these indications (such as PNH), it is important to emphasize that C5 inhibition provides only symptomatic relief. Over the past decade there has been an impressive increase in our comprehension...”

Reviewer 2 Report

The manuscript by Bernuy-Guevara et al. entitled “The inhibition of complement system in formal and emerging indications: Results from parallel one-stage pairwise and network meta-analyses of clinical trials and real-life data studies (reference biomedicines-892112)” reports on how efficacious are C5 inhibitors and C1 esterase inhibitors in the treatment of PNH, aHUS and  rgMG plus treatment/prevention of aAMR and DGF in kidney transplant. A systematic review is executed for this topic. Stronger evidence on beneficial effects in PNH, aHUS and rgMG, as well as in aAMR in kidney transplants, is currently available for C5 inhibitors as compared to C1 esterase inhibitors. The authors performed an extensive study and describe interesting features for the clinical field. These studies are required to make the final (best) choice of the complement inhibitor in specific patient populations now and in the future. Next issues need to be addressed:

1.)Is there an effect of the genetic background of the included patients on the outcome described in this manuscript?

2.)Nowadays because of costs issues many studies have been executed and described in patients cohorts where drug administration is tapered or stopped after a certain period of time. Have these studies been included in the current data set?

3.)Line 214 and 215: “The summary estimates were affected by heterogeneity.......”. Please elaborate on this item.

4.)No effect of the C1 esterase inhibitor for treating aAMR was established. Is this because of this particular patient group? Can you elaborate on this?

5.)For all indications studied in the present manuscript the C5 complement inhibitors need to be administered life long?

Advise: Minor revision.

Author Response

Reviewer 2

The manuscript by Bernuy-Guevara et al. entitled “The inhibition of complement system in formal and emerging indications: Results from parallel one-stage pairwise and network meta-analyses of clinical trials and real-life data studies (reference biomedicines-892112)” reports on how efficacious are C5 inhibitors and C1 esterase inhibitors in the treatment of PNH, aHUS and  rgMG plus treatment/prevention of aAMR and DGF in kidney transplant. A systematic review is executed for this topic. Stronger evidence on beneficial effects in PNH, aHUS and rgMG, as well as in aAMR in kidney transplants, is currently available for C5 inhibitors as compared to C1 esterase inhibitors. The authors performed an extensive study and describe interesting features for the clinical field. These studies are required to make the final (best) choice of the complement inhibitor in specific patient populations now and in the future. Next issues need to be addressed:

1) Is there an effect of the genetic background of the included patients on the outcome described in this manuscript?

We thank you for this important comment. The findings presented here are restricted to answer the main review question, that is how complement inhibitors are efficacious to treat or prevent various conditions associated to complement activation. Nevertheless, the genetic background should be considered as an explanation for heterogeneity in calculations. According to this, the text has been modified at the end the third paragraph in subsection “Important message” of the Discussion, lines 245 to 249, and now reads:

“This effect persists in RCT and also in real-life NRSI, despite the fact that the quality of the included individual studies greatly differ, and the existence of statistical heterogeneity and reporting bias in many studies. In particular, with regards to differences between participants in different latitudes leading to heterogeneity, the genetic background should be considered, among other factors.”

2) Nowadays because of costs issues many studies have been executed and described in patients cohorts where drug administration is tapered or stopped after a certain period of time. Have these studies been included in the current data set?

In addition to clinical trials, calculations on all available observational studies that may be considered real-life (or real-world) evidence on the use of complement inhibitors are presented. Indeed, as comparators, patients who stopped complement inhibitors (off-treatment patients) represented almost a third of the comparison in such real-life studies. These individuals were clearly distinguished from those who never underwent complement inhibition (pre-eculizumab era) and those before receiving complement inhibitors (pre-treatment state).

Please see at the eighth paragraph in subsection “Systematic narrative synthesis” of the Results section, lines 129 to 133, the text now reads:

“The comparators in real-life NRSI were pre-eculizumab era individuals, i.e., patients who never underwent complement inhibition (56%), and individuals treated with complement inhibitors in their off-treatment state, i.e., patients who discontinued complement inhibitors for various reasons (33%), and patients in their pre-treatment state, i.e., before receiving complement inhibition (11%).”

3)Line 214 and 215: “The summary estimates were affected by heterogeneity.......”. Please elaborate on this item.

We thank you for pointing this issue. Heterogeneity particularly affected the subgroup analysis assessing eculizumab to prevent acute antibody-mediated rejection, as funnel plot asymmetry was noted in the analysis of the effect of complement inhibition as prevention of delayed graft function.

Please see at the seventh paragraph in subsection “Quantitative analysis” in the Results section, lines 222 to 224, the text now reads:

“Heterogeneity (43%) affected summary estimates in subgroup analysis of eculizumab to prevent aAMR, and funnel plot asymmetry was particularly noted in the analysis on prevention of DGF.”

4) No effect of the C1 esterase inhibitor for treating aAMR was established. Is this because of this particular patient group? Can you elaborate on this?

Although the image from this analysis is not presented in the manuscript nor in the supplementary material, no effect from C1 esterase inhibitors to treat acute antibody-mediated rejection was observed.

For the information of the Reviewer, the forest plot of this analysis at the pairwise level involving C1 esterase inhibitors berinert and cinryse is show as following:

Verification of an effect from other molecules is ongoing and belong to an update of the meta-analysis presented here.

5) For all indications studied in the present manuscript the C5 complement inhibitors need to be administered life long?

We thank you for this relevant comment. Authorization reports do not address specifically this aspect of the therapy using these molecules. Our meta-analysis is also not able to answer to this question. Probably, more real-world evidence is necessary. In an update of this meta-analysis this issue will be probably solved. To the moment, our finding show that C5 inhibition is efficacious and that the two C5 inhibitors eculizumab and ravulizumab are similar regarding their efficacy (at the moment only considering paroxysmal nocturnal hemoglobinuria and atypical hemolytic uremic syndrome).

Please see at the fourth paragraph in subsection “Important message” in the Discussion section, lines 253 to 256, the text now reads:

“In our opinion, decisions to use one or the other C5 inhibitor should be based on other factors such as costs, insurance coverage, availability, local expertise, etc. In addition, in these and other indications long term use of complement inhibitors remains to be evaluated.”

Reviewer 3 Report

There are no specific comments for the authors.

Author Response

There are no specific comments for the authors.

We thank you for reading of our manuscript.
